# Mechanical Performance of Wool-Reinforced Epoxy Composites: Tensile, Flexural, Compressive, and Impact Analysis

**DOI:** 10.3390/ma18235391

**Published:** 2025-11-29

**Authors:** Carlos Ruiz-Díaz, Guillermo Guerrero-Vacas, Óscar Rodríguez-Alabanda

**Affiliations:** Department of Mechanical Engineering, University of Córdoba, 14071 Córdoba, Spain; guillermo.guerrero@uco.es (G.G.-V.); orodriguez@uco.es (Ó.R.-A.)

**Keywords:** natural-fibre composites, sheep wool fibres, epoxy matrix, mechanical properties, tensile stress, flexural behaviour, compressive stress, sustainable materials

## Abstract

This study situates washed sheep-wool fibres as a sustainable reinforcement candidate for epoxy matrices and evaluates their mechanical response under tensile, flexural, compressive, and Charpy impact loading. The objective of this work is to assess whether short, washed sheep-wool fibres can function as a sustainable reinforcement for epoxy matrices, and to identify optimal fibre length–content windows that improve mechanical behaviour for engineering applications. Moulded–machined specimens were produced with fibre lengths of 3, 6, and 10 mm and contents of 1.0–5.0 wt.%, depending on the test; neat epoxy served as the reference. In tension, selected formulations—particularly 10 mm/1.5 wt.%—showed simultaneous increases in ultimate stress and modulus relative to the neat resin, corresponding to gains of about 10% in ultimate tensile stress and 50% in tensile modulus, at the expense of ductility. In flexure, the modulus decreases by roughly 15–35% compared with the matrix, whereas configurations with 3–6 mm at 2.5–5 wt.% raise the fracture stress by about 35–45% and improve post-peak resistance. In compression, reinforcement markedly elevates yield stress, with increases of up to about 160% at 3 mm/2 wt.%, while the ultimate strain decreases moderately. In Charpy impact, all reinforced materials underperform the resin, with absorbed energy reduced by roughly 75–93% depending on fibre length and content, with 3 mm/1 wt.% being the least affected. A two-factor analysis of variance (ANOVA) indicates that fibre length primarily governs tensile and compressive behaviour, while fibre content dominates flexural and impact responses. Overall, the findings support wool fibres as a viable reinforcement when length and content are optimized, pointing to their use in non-structural to semi-structural industrial components such as interior panels, housings, casings, protective covers, and other parts where moderate tensile/compressive performance is sufficient and material sustainability is prioritised.

## 1. Introduction

Wool, a natural product obtained from sheep shearing, has long been valued for its thermal and insulating properties. However, the evolution of the textile industry and the rise of synthetic fibres have drastically reduced the demand for low-grade wool, creating environmental and economic burdens due to stockpiling and disposal in producer regions [1,2,3]. Most of the flocks that generate this coarse, heterogeneous wool are primarily farmed for meat, milk, or grazing management rather than for fibre, but the animals must still be shorn annually for health and welfare reasons, so low-grade wool continues to be produced as an unavoidable by-product with little or no market value. This situation has prompted efforts to valorise wool residues with minimal processing and reduced ecological footprint, and this motivates the present study on their use as reinforcements in epoxy composites.

Agricultural applications have drawn particular attention: unprocessed wool can act as a slow-release nitrogen fertilizer, and wool-based mats, pellets, and soil conditioners have shown promise in horticultural and forestry uses [4,5]. EU-funded initiatives (e.g., LIFE GREENWOOLF) have further demonstrated the feasibility of transforming low-quality wool waste into biodegradable fertilizers and mulching materials, aligning with circular-economy goals [6]. Despite these advances, the agricultural route alone cannot absorb the yearly surplus of low-grade wool, motivating the exploration of alternative, higher-value material uses [7,8].

Within materials engineering, natural fibres (e.g., flax, jute, hemp, sisal) have been widely investigated as polymer reinforcements due to their low cost, low density, renewability, and favourable life-cycle metrics, while maintaining competitive mechanical performance in selected applications [9,10]. Epoxy resins, in turn, are staple thermosets for advanced composite structures because of their strong adhesion, chemical resistance, and thermal stability, and they are widely used both as matrix resins and as structural adhesives in bonded, co-bonded, and co-cured composite joints in aerospace, automotive, and marine applications [11,12,13,14]. Combining epoxy matrices with natural fibres has been proposed as a route towards composites with higher renewable content and improved life-cycle metrics, even though conventional epoxy matrices are not biopolymers and are not biodegradable [15]. However, the performance of an epoxy composite depends sensitively on fibre–matrix interactions, load transfer, fibre length, and fibre content, which warrant systematic study [16].

Compared with plant fibres, the structural use of animal fibres—particularly sheep wool—in polymer composites remains underexplored. Prior work on wool has focused more on thermal/acoustic insulation than on structural reinforcement in epoxy matrices [17,18]. Nonetheless, wool’s intrinsic elasticity and energy-absorption capacity suggest potential in non-structural to semi-structural components, provided that the fibre geometry and content are judiciously optimized [19].

At the same time, the literature presents diverging views on key issues relevant to wool–epoxy systems. First, short-fibre micromechanics indicates that tensile stress gains hinge on effective load transfer and a critical fibre length, which may be harder to achieve with lower-modulus animal fibres [20,21]. Second, the impact response of natural-fibre composites varies widely: some studies report energy-absorption benefits, whereas others show degradation depending on fibre length, content, and processing-induced defects [22,23]. Third, opinions differ on whether washed (untreated) wool is sufficient for adhesion and moisture resistance or whether surface treatments are needed [24]. These divergences underscore the need for controlled, epoxy-based studies isolating the effects of fibre length and fibre content. To the best of the authors’ knowledge, there is no available study on short-fibre-reinforced epoxy composites that systematically covers fibre lengths of 3, 6, and 10 mm combined with contents of 1.0–5.0 wt.% under a unified experimental design. This lack of directly comparable data underscores the need for reference mechanical results in this length–content window to benchmark wool–epoxy systems against other short-fibre composites.

This work investigates the feasibility of using washed sheep wool fibres as reinforcements in an epoxy matrix by systematically varying the fibre length (3, 6, and 10 mm) and content (1.0–5.0 wt.%) and benchmarking against neat epoxy under standardized procedures. By disentangling the roles of length and content across tensile, flexural, compressive, and Charpy impact loading, this study provides actionable guidance for designing wool-reinforced epoxies while contributing to the circular-economy valorisation of low-grade wool. In brief, our results show that selected wool configurations can increase tensile stress/modulus and compressive yield stress relative to neat epoxy, whereas flexural modulus and impact performance are more sensitive to fibre content—clarifying when and how wool can function as an effective, sustainable reinforcement.

## 2. Materials and Methods

### 2.1. Materials

The polymer matrix was a commercial, solvent-free, two-component epoxy system, DIPOXY-2K-700 (Dipoxy International GmbH, Hanau, Germany), mixed at a 2:1 resin–hardener mass ratio, self-degassing, transparent, and suitable for cast layers of up to 10 mm. The processing time ranges from 10 to 60 min depending on temperature, and full cure is achieved within 2–14 days, following the manufacturer’s guidance.

The natural reinforcement consisted of coarse wool fibres from Segureña sheep (Wool Dreamers, Cuenca, Spain), a local breed from southeastern Iberia, with medium-to-coarse texture and of non-textile grade. The fibres were subjected to a basic washing to remove superficial dirt without chemical surface treatment, and then they were used as dispersed short fibres with controlled length and weight fraction. Optical microscopy (Leica Microsystems S.L.U., L’Hospitalet de Llobregat, Spain) confirmed heterogeneity in diameter (≈ 20–70 μm overall; means ≈ 40–45 μm) and minor residues (straw/soil; low). Figure 1 shows the characterization of the wool obtained by optical microscopy.

Figure 1a–d show the heterogeneity of the fibres in thickness and arrangement, with diameters displaying significant variation and surface irregularities typical of unprocessed natural materials. Table 1 complements these qualitative observations with morphological and visual characterization, including colour, type, and presence of residues, along with the maximum, minimum, and average diameters measured in the samples.

For mould fabrication, EN AW-2024-T6 aluminium was used. To aid in demoulding and protect cavity surfaces, an fluorinated ethylene propylene (FEP) based release coating (Elite 8840, Tecnimacor S.L., Córdoba, Spain) was applied at 60–70 μm thickness.

### 2.2. Equipment

Moulds for tensile and flexural specimens were machined on an FTV-1 manual vertical mill (Lagun Machine Tools S.L.U., Azkoitia, Gipuzkoa, Spain). Compression and impact moulds, requiring higher precision and minimal vibration, were produced on a CNC vertical machining centre QP2026-L (Falcon Machine Tools Co., Ltd., Changhua, Taiwan) equipped with FANUC control, a high-performance spindle, linear guides, and travels of 660 mm × 520 mm × 508 mm (X, Y, Z). Dimensional finishing after moulding was carried out on a manual horizontal lathe SP/165 (Pinacho, Zaragoza, Spain).

Fibre characterization, fracture inspection, and specimen preparation used a DVM6-A digital optical microscope (Leica Microsystems S.L.U., L’Hospitalet de Llobregat, Spain) with a motorized focus, XY stage, and a CMOS sensor up to 10 MP and 37 fps. For fracture-surface inspection, the microscope was operated in reflected-light mode with an FOV 43.75 objective (nominal NA = 0.007). Images were acquired at 1600 × 1200 px (8-bit RGB), corresponding to a field of view of approximately 34.5 mm × 25.8 mm and a spatial sampling of ≈21–22 μm per pixel. The same optical configuration was used for all fracture surfaces reported in this study. Resin and wool were weighed on a WLC 2/A2 precision balance (Radwag, Toruń, Poland; 0.01 g resolution, 2 kg capacity, tare), and specimen dimensions were verified with a digital calliper (Mitutoyo Corporation, Kawasaki, Japan; 0.01 mm).

Thermal curing under a controlled temperature was performed in a 2,000,210 laboratory oven (J.P. Selecta S.A., Abrera, Barcelona, Spain; 1200 W, 0–250 °C).

Mechanical testing used a universal testing machine M-405 (Servosis S.L., Madrid, Spain; 1–500 kN, max 180 mm/min, 1200 mm crosshead travel) with tensile grips, a three-point bending fixture, and flat compression platens, while Charpy impact toughness was measured with an RKP 450 pendulum (ZwickRoell GmbH & Co. KG, Ulm, Germany; 450 J, −45 to +85 °C).

### 2.3. Methods

Moulded–machined specimens were fabricated to evaluate the influence of fibre length and wool mass fraction in an epoxy matrix. The experimental design was conceived to ensure traceability and reproducibility

Standard specimens for tensile, flexural, compressive, and Charpy impact tests were produced by gravity-casting of wool–epoxy mixtures into aluminium moulds, followed by machining to the final geometries specified in Table 2. The two epoxy components were weighed at a 2:1 resin–hardener mass ratio and hand-mixed for approximately 5 min. Washed Segureña wool was cut to nominal fibre lengths of 3, 6, or 10 mm and weighed to reach the target mass fractions (1.0–5.0 wt.%). For each formulation, the required amount of wool was gradually incorporated into the fresh resin under manual stirring until a visually homogeneous suspension of short, randomly oriented fibres was obtained (no laminates or stacking sequence were used). The mixtures were then poured by gravity into FEP-coated aluminium moulds: flat plate cavities for tensile, flexural, and Charpy bars, and cylindrical cavities for compression specimens. To limit air entrapment, mixing was performed manually at low speed, and the moulds were filled slowly from one side, allowing bubbles to rise to the free surface before gelation. No additional vacuum-degassing step was applied in this study. After initial gelation at room temperature, the filled moulds were cured in a ventilated oven at 60 °C for 24 h, demoulded, and finally machined to the standardized dimensions listed in Table 2.

Figure 2 shows the mechanised moulds designed for the different mechanical tests carried out in this study.

All mechanical tests—tensile, flexural, compressive, and Charpy impact—were performed in accordance with the corresponding UNE-EN ISO standards. Specimens were prepared as previously described in the Section 2.1 and Section 2.2 and subsequently machined to meet the required dimensions (see Table 2). The tensile tests followed UNE-EN ISO 527-1 [25] and 527-4 [26], using Type 2 specimens of 250 mm × 25 mm × 4 mm(length × width × thickness) (n = 3 per configuration). Tests were carried out on a Servosis M-405 universal testing machine with a gauge length of 150 ± 1 mm between grips and a crosshead speed of 0.20 mm/s, with the lower grip fixed. The parameters recorded were yield stress (*σ*ᵧ*_,t_*), ultimate stress (*σ_u,t_*), fracture stress and strain (*σ_b,t_*, *ε_b,t_*), and elastic modulus (*E_t_*). The tensile modulus *E_t_* was computed by linear regression over 0.05–0.25% strain on the initial linear portion of the stress–strain curve; machine/grip compliance was checked and found to be negligible for the gauge length used. Flexural tests were conducted according to UNE-EN ISO 178 [27] on prismatic specimens of 100 mm × 10 mm × 4 mm (n = 5 per configuration) in a three-point bending setup, using 10 mm rollers and a support span of 64 mm. The load was applied at a controlled speed until fracture, and the ultimate flexural stress (*σ_u,f_*), flexural modulus (*E_f_*), and fracture stress (*σ_b,f_*) were determined. Compression tests were performed in line with UNE-EN ISO 604 [28] using cylindrical specimens (Ø20 × 25 mm; n = 5 per configuration), with machined ends to ensure parallelism. Testing was conducted on the Servosis M-405 with careful axial alignment, and stress–strain curves were used to determine the compressive yield stress (*σ_y,c_*), ultimate stress (*σ_u,c_*), yield strain (*ε_y,c_*), and ultimate strain (*ε_u,c_*). In compression, the ultimate and fracture values coincide and are reported interchangeably. The Charpy impact tests followed UNE-EN ISO 179 [29], using prismatic specimens of 10 mm × 4 mm × 80 mm (height x thickness x length) (n = 5 per configuration). Specimens were V-notched (radius r = 0.25 mm) and tested with the notch facing the striker. Tests were conducted on a ZwickRoell RKP 450 pendulum (450 J, −45 to +85 °C), and the effective absorbed energy (*W_ef_*) was obtained from the loss of potential energy after fracture.

Fracture morphology was then examined to relate the observed surfaces with the mechanical behaviour reported in the different tests. The analysis aimed to identify the mechanisms governing failure in neat epoxy and wool–epoxy composites, and to explain the role of fibre length and content in crack initiation, propagation, and energy dissipation. To reinforce interpretation and ensure statistical validity, an ANOVA under the principal effects approach (PEVA) was performed using Minitab 21.2.0 (Minitab LLC, State College, USA). Fibre length and mass fraction were considered as main factors, and their effects were evaluated on tensile, flexural, compressive, and impact performance. Normality (Anderson–Darling) and homoscedasticity (Levene) were verified at α = 0.05 prior to two-factor ANOVA. In the model summaries, S denotes the standard error of regression. To ensure comparability and focus on the ultimate state, the response variable was the ultimate stress (*σ_u_*) for tensile, flexural, and compressive tests, while for Charpy the effective absorbed energy was used. The analysis included main effect plots and considered interaction terms (L × W), which were reported when significant.

## 3. Results and Discussion

### 3.1. Tensile Test Results and Analysis

The tensile behaviour of epoxy–wool composites was examined as a function of fibre length (3, 6, and 10 mm) and weight content (1.5 and 3.0 wt.%). The yield stress (*σ_y,t_*), ultimate and break stresses (*σ_u,t_*, *σ_b,t_*), elastic modulus (*E_t_*), and strain at break (*ε_b,t_*) were determined, reported as the mean ± standard deviation, and compared with the neat resin. Table 3 summarizes *E_t_* and *ε_b,t_* for each fibre length and content, providing a direct reference for the stiffness and ductility of each formulation.

The incorporation of fibres alters both the stiffness and deformability of the composites. While the elastic modulus shows notable increases in some cases, the strain at break generally decreases compared to the neat resin. Figure 3 complements this by showing the evolution of the characteristic tensile stresses as a function of fibre length and content.

As shown in Table 3 and Figure 3, wool reinforcement only produces a clear increase in stiffness (*E_t_*) at 10 mm/1.5 wt.%, where the modulus reaches 490.4 ± 101.5 MPa versus 331.0 ± 39.2 MPa for the resin, i.e., an increase of approximately 50% in tensile modulus with respect to the neat epoxy. Other formulations remain similar or lower (e.g., 269.2 ± 72.7 MPa at 6 mm/1.5 wt.%, 238.0 ± 4.1 MPa at 6 mm/3.0 wt.%, 282.7 ± 20.6 MPa at 3 mm/3.0 wt.%, and 251.0 ± 27.3 MPa at 10 mm/3.0 wt.%). For ultimate stress, the maximum is ≈27 MPa at 10 mm/1.5 wt.%, about 10% higher than the ultimate tensile stress of the neat resin, whereas 3 mm and 6 mm remain at ≈20–22 MPa and the matrix at ≈24–25 MPa; increasing to 3.0 wt.% yields no improvement, even at 10 mm. Meanwhile, *ε_b,t_* decreases from 5.65 ± 1.33% (unreinforced) to ≤ 0.34%, reaching 0.00% at 3 mm/3.0 wt.%. Overall, the results indicate that exceeding the critical fibre length—here, 10 mm within the tested range—and keeping moderate fractions (1.5 wt.%) will optimize *E_t_* and *σ_u,t_*, whereas higher loadings are penalized by aggregation and interfacial defects. This is consistent with the findings of Patrucco et al. [18]—who highlighted fibre–matrix adhesion and warned about agglomeration at high loadings in wool–polymer composites—and with recent studies placing optimal short-fibre lengths between ~5 and 30 mm, depending on system and processing (e.g., 5 mm in jute/epoxy preform; 10–30 mm in sisal/PP; 30 mm in areca/epoxy), supporting the effectiveness of 10 mm as a “moderate length” [22,24].

It is worth noting that, for the shortest fibre length at the lowest content (3 mm/1.5 wt.%), the ultimate tensile stress decreases by about 17.5% compared with the neat epoxy. This behaviour is consistent with a defect-dominated regime in which fibres that are below or close to the effective load-bearing length do not contribute significantly to tensile reinforcement. Instead, they primarily act as additional stress concentrators (fibre ends, local stiffness mismatches, voids, and interfacial defects), promoting earlier crack initiation and limiting the potential for fibre bridging and pull-out to increase the load-carrying capacity.

### 3.2. Flexural Test Results and Analysis

This section examines the flexural behaviour of epoxy–wool composites with fibre lengths of 3, 6, and 10 mm at 2.5 and 5.0 wt.%. The analysis focuses on the flexural modulus (*E_f_*) and the load-bearing capacity at peak and fracture (*σ_u,f_*, *σ_b,f_*), derived from load–deflection curves. The results are presented as the mean ± standard deviation relative to the neat resin. Table 4 summarizes the flexural modulus.

Table 4 shows the effect of reinforcement on the flexural elastic modulus, varying with fibre length and content. To complement stiffness with load-bearing capacity, Figure 4 reports the characteristic flexural stresses.

Table 4 confirms a reduction in flexural modulus compared to the neat resin (1961.3 ± 134.5 MPa), with composites ranging from 1237.2 to 1672.9 MPa, i.e., a decrease of approximately 15–35% relative to the neat epoxy. The highest *E_f_* appears in 3 mm/5.0 wt.% (1672.9 ± 45.4 MPa) and 6 mm/2.5–5.0 wt.%, whereas the lowest is found at 10 mm/5.0 wt.% (1237.2 ± 149.9 MPa). This suggests that short–medium lengths and moderate fractions better preserve stiffness than longer, heavily loaded combinations. Figure 4 shows that the flexural stress of the matrix is 60.8 MPa, while most composites fall between 45.0 and 57.0 MPa, consistent with reductions in peak flexural stress in the order of 15–25%, with peaks again at 3–6 mm/5.0 wt.%. In contrast, the fracture stress exceeds the resin baseline (33.3 MPa) in several cases (45.0–48.0 MPa), corresponding to increases of roughly 35–45%, indicating post-peak load-bearing despite reductions in peak stress and modulus.

These trends align with studies on natural-fibre composites, which identify effective windows of fibre length and content, whereas excessive values promote agglomeration and poor load transfer. In wool–epoxy, Patrucco et al. [18] emphasize weak interfacial bonding with thermosets and limited property gains without treatments (e.g., alkaline/silane), explaining why our 10 mm/5.0 wt.% formulations did not improve *E_f_* or *σ_u,f_*. The enhancement of *σ_b,f_* is consistent with extrinsic mechanisms such as fibre bridging and pull-out, which sustain load beyond the peak. As detailed by Lubineau et al. [30], these mechanisms increase post-peak toughness when interfacial traction is effective—consistent with our 3–6 mm results and contrasting with those for 10 mm/5.0 wt.% (more decohesion/porosity). Recent reviews [31] similarly conclude that mechanical optima occur at moderate lengths and fractions, with fibre treatments further improving load transfer and flexural performance, mirroring the behaviour observed here.

For flexural loading, the shortest wool fibres also lead to a reduction in strength. The incorporation of 3 mm fibres produces decreases in flexural strength of up to approximately 25% relative to the neat resin, depending on fibre content, as illustrated in Figure 4. As in tension, this behaviour is consistent with a defect-dominated regime, where very short fibres mainly introduce additional stress concentrators (fibre ends, local stiffness mismatches, voids, and interfacial decohesion) in the tensile zone of the bent specimens, instead of acting as effective load-bearing bridges. This interpretation is consistent with the fracture morphologies observed around short fibres (Section 3.5).

### 3.3. Compressive Test Results and Analysis

This section analyses the compressive response of epoxy–wool composites with fibre lengths of 3, 6, and 10 mm and fractions of 1.0 and 2.0 wt.%. From the σ–ε curves, characteristic yield and ultimate parameters were determined and are reported as the mean ± standard deviation relative to the unreinforced resin. In this case, the maximum stress coincides with the ultimate state, so no distinction is made between the maximum and fracture stresses. For an integrated view, Table 5 summarizes the strains at yield (*ε_y,c_*) and at fracture (*ε_b,c_*).

Table 5 shows an increase in compressive yield strain in the composites compared to the resin, accompanied by a slight reduction in fracture strain, with variations depending on fibre length and content. To complement these strain data, Figure 5 presents the compressive stresses—*σ_y,c_* and *σ_u,c_*—as a function of the different formulations.

Table 5 shows that the compressive yield strain increases in all composites compared to the neat resin (6.6 ± 0.5%), reaching ≈ 9–11% depending on fibre length and content. In parallel, the fracture strain decreases from 62.8 ± 1.8% in the matrix to ≈ 42–52% in the reinforced systems. Figure 5 confirms that the compressive yield stress rises from 20.2 MPa in the resin to ≈ 41.8–52.7 MPa in the composites, with a maximum at 3 mm/2 wt.% (52.7 MPa). A clear trend with fibre length is observed (≈ 3 mm > 6 mm > 10 mm), along with a length-dependent effect of fibre content: at 3 mm, increasing from 1 to 2 wt.% enhances *σ_y,c_*; at 6–10 mm, higher content tends to provide no improvement or even reduce *σ_y,c_*. Overall, wool reinforcement raises the elastic threshold but shortens the ultimate strain, i.e., greater resistance to yielding but lower final ductility. This pattern is consistent with the micromechanics of polymer composites under compression: fibres constrain matrix shear and delay flow, but failure is governed by local instabilities (microbuckling, kink bands) and by the heterogeneity inherent to discontinuous reinforcements (fibre ends, voids, length dispersion). In recent scientific reviews, Islam et al. [24] describe how kink bands emerge early and reduce ductility, while Larsson et al. [32] document the sensitivity to local defects under compression—two mechanisms that explain the combination of higher *σ_y,c_* with lower *ε_b,c_* observed here. The best performance at 3 mm/2 wt.% suggests an effective short-fibre window, close to the critical fibre length, where load transfer is most efficient. Moving away from this window—by increasing the fibre length to 6–10 mm or raising the content at these lengths—elevates the defect density and geometric inefficiency, resulting in reduced *ε_y,c_* and shortened post-yield strain. This interpretation aligns with the classical micromechanics of discontinuous reinforcements, as described by the Kelly–Tyson model and its subsequent developments [21], and with later syntheses by Thomason [33] emphasizing the importance of effective fibre length and its dispersion in short-fibre composites. Similarly, reviews on epoxy reinforced with natural fibres [34] stress that optimal properties are achieved in moderate ranges of fibre length and content, whereas further increases penalize ductility and compressive performance, in agreement with our findings.

### 3.4. Charpy Impact Test Results and Analysis

Figure 6 presents the Charpy effective energy absorption (*W_ef_*) of the epoxy–wool composites at different fibre lengths and contents, relative to the unreinforced resin.

In notched Charpy tests, neat epoxy absorbs *W_ef_* = 1.30 J. With short wool fibres, the energy drops to 0.33 J (3 mm/1 wt.%), 0.23 J (6 mm/1 wt.%), and 0.19 J (10 mm/1 wt.%); at 2 wt.%, it decreases further to 0.16 J, 0.10 J, and 0.09 J, respectively. Thus, the composites retain only ~7–25% of the matrix energy, with 3 mm/1 wt.% showing the best performance. A clear trend emerges: at fixed content, *W_ef_* decreases with fibre length, and at fixed length, it is higher at 1 wt.% than at 2 wt.%. This behaviour reflects short-fibre composites where interfacial decohesion dominates and the effective length is below or near the critical length, leading to more crack initiators at fibre ends and pores and less dissipative pull-out. The results align with recent reviews on natural-fibre composites and studies relating impact response to fibre length distribution and load transfer efficiency [35,36]. A micromechanical interpretation based on Kelly–Tyson, extended for very short fibres, further supports the idea that, outside an optimal window, higher length or content increases the interfacial area and notch sensitivity without improving load transfer [34,37].

### 3.5. Fracture Analysis and Visual Inspection

This section relates the fracture morphology observed in the specimens to the mechanical results previously reported in tension, flexure, compression, and Charpy impact. The objective is to clarify how differences between formulations (fibre length and content) originate. For this purpose, the fracture surfaces of neat epoxy are compared with those of epoxy–wool composites, using representative micro/macrographs (Figure 7, Figure 8, Figure 9 and Figure 10) and reference values from earlier tables and figures. Fractographic observations were performed with a Leica DVM6-A digital microscope operated in reflected-light mode under the optical configuration described in Section 2.2 (FOV 43.75 objective, 1600 × 1200 px, spatial sampling ≈ 21–22 μm/px), which in this study allowed us to visually distinguish individual wool fibres (≈20–40 μm in diameter) and to qualitatively identify their failure mode (predominantly fibre pull-out rather than fibre breakage).

The fracture surfaces reveal distinct mechanisms depending on reinforcement and formulation: (a) neat resin shows a smooth, glossy plane typical of brittle cleavage; (b–e) with 3–6 mm fibres, rougher surfaces with voids, fibre ends, and interfacial debonding appear, evidencing pull-out and matrix micro-shear; (f) at 10 mm/1.5 wt.%, the surface is highly tortuous, with long fibres pulled out or bridging the crack, consistent with the maximum tensile stress (≈27 MPa) despite low ductility; (g) at 10 mm/3.0 wt.%, smooth zones alternate with agglomerates and pores, dominated by debonding and short pull-out, explaining the drop in modulus and tensile stress compared with 10 mm/1.5 wt.% (saturation and defect effects). Overall, the transition from “smooth surface → rough with pull-out/bridging” and the contrast between 10 mm/1.5 wt.% (bridging) and 10 mm/3.0 wt.% (defects) account for the slight stiffness increase in the former and the penalty at higher fibre fractions.

In all reinforced tensile specimens, most wool fibres fail by interfacial debonding followed by pull-out, whereas clean fibre breakage is comparatively rare. This pull-out-dominated fracture mode is consistent with the modest increase in ultimate tensile stress and the severe loss of ductility reported in Section 3.1.

The fracture surfaces after flexural testing show a clear transition between resin and composites: (a) neat resin presents a smooth, glossy plane characteristic of brittle fracture with minimal crack deviation; (b–c) 3 mm/2.5–5.0 wt.% displays rougher surfaces with numerous pull-out marks and microcavitation, evidencing short pull-out and moderate crack tortuosity, which indicate additional energy dissipation; (d–e) 6 mm/2.5–5.0 wt.% reveals longer pull-out lengths and fibre bridging zones, consistent with the formulations that best preserve flexural modulus; (f) 10 mm/2.5 wt.% shows a highly tortuous surface with long fibres partially extracted, suggesting a bridging contribution despite local heterogeneity; (g) 10 mm/5.0 wt.% alternates smooth patches with agglomerates and voids, where interfacial debonding and short pull-out dominate, consistent with the drop in *E_f_* for this higher fibre loading. Overall, the transition from “smooth → rough with pull-out/bridging” when moving from resin to composites, together with the contrast between short/medium fibres and 10 mm/5.0 wt.%, accounts for the superior performance of intermediate formulations and the degradation observed when both fibre length and content increase simultaneously.

As in tension, the dominant fracture mechanisms are fibre pull-out and interfacial debonding, rather than complete fibre rupture, so flexural energy dissipation is mainly governed by crack deflection around partially debonded fibres.

Observations of the fracture surfaces reveal how fibre reinforcement modifies the failure behaviour of the resin under compression. (a) Neat resin exhibits a smooth surface with nearly straight axial splits and slight barrelling, typical of brittle cleavage under compression; the crack propagates with little deviation and without extensive crushing zones; (b) 3 mm/1 wt.% shows a rougher surface with oblique shear bands, local crushing, and cavities, together with debonding marks and short fibre pull-out; this microstructure suggests that the fibres constrain matrix shear, raise the yielding threshold, and promote multiple cracks rather than a single fracture plane; (c) 3 mm/2 wt.% presents a more tortuous surface with extended crushing, porosity/agglomerates, and evident pull-out; micro-buckling/kink planes are observed coalescing into a predominant axial crack. The reinforcement increases resistance to flow but concentrates instability in local zones, consistent with the lower final ductility. Overall, the transition from neat resin to composites is from clean axial splitting to crushing- and shear-dominated surfaces with debonding and fibre pull-out, which explains the increase in compressive yield stress and the reduction in compressive strain at break: greater ability to resist initial yielding, but fracture governed by local instabilities.

Fracture surface analysis reveals how fibre reinforcement modifies resin failure. In Figure 10(a1,a2), neat resin displays smooth, shiny surfaces with minimal chipping, typical of brittle, notch-controlled fracture; cracks propagate almost straight, with little deviation. In Figure 10(b1,b2), 3 mm/1 wt.% exhibits increased roughness, small cavities, shear planes, interfacial debonding, and short fibre pull-out, indicating crack deflection and energy dissipation, consistent with the highest *W_ef_* among the reinforced samples. In Figure 10(c1,c2), 3 mm/2 wt.% shows local plastic deformation, porosity, agglomerates, and flattened regions, with multiple crack initiations at defects and fibre ends; debonding and short pull-out dominate, but higher defect density and notch sensitivity reduce the dissipative contribution. Overall, the transition from neat resin to reinforced composites progresses from clean cleavage to tortuous, fibre–matrix-interacting fracture surfaces, explaining both the decrease in *W_ef_* with increased fibre content (3 mm/1 wt.% > 3 mm/2 wt.%) and the general trend in impact resistance across lengths and fibre loadings.

### 3.6. Statistical Analysis

In order to reinforce the experimental conclusions and quantitatively evaluate the influence of design parameters, a statistical analysis was performed on the main results obtained from the mechanical tests. Specifically, a two-factor ANOVA was applied (reinforced subset: 3, 6, and 10 mm × 1.0 to 5.0 wt.%, excluding unreinforced specimens) to study the effects of fibre length (L, mm) and wool content (W, wt.%) on the selected response of each test: tension (*σ_u,t_*), flexural (*σ_u,f_*), compression (*σ_u,c_*), and impact (*W_ef_*). Additionally, main effects plots were generated to visualize the magnitude and individual impact of each factor. This analysis allows for the identification of statistically significant trends and provides an objective criterion for comparing the behaviour of the different formulations. Table 6 shows the results of the analysis of variance for the indicated variables.

The model summary for the selected results provides the data included in Table 7.

The regression equations fitted for the selected response variables are presented below (Equations (1)–(4)).
**Response****Equation**
*σ_u,t_*22.210 − 1.265·I_L = 3_ − 1.022·I_L = 6_ + 2.287·I_L = 10_ + 0.703·I_W = 1.5_ − 0.703·I_W = 3.0_ − 1.465·I_L = 3_·I_W = 1.5_ + 1.465·I_L = 3_·I_W = 3.0_ −0.282·I_L = 6_·I_W = 1.5_ + 0.282·I_L = 6_·I_W = 3.0_ + 1.747·I_L = 10_·I_W = 1.5_ − 1.747·I_L = 10_·I_W = 3.0_.(1)*σ_u,f_*37.86 + 0.02·I_L = 3_ + 2.37·I_L = 6_ − 2.40·I_L = 10_ − 2.36·I_W = 2.5_ + 2.36·I_W = 5.0_ − 7.82·I_L = 3_·I_W = 2.5_ + 7.82·I_L = 3_·I_W = 5.0_ + 7.14·I_L = 6_·I_W = 2.5_ − 7.14·I_L = 6_·I_W = 5.0_ + 0.68·I_L = 10_·I_W = 2.5_ − 0.68·I_L = 10_·I_W = 5.0_(2)*σ_u,c_*68.256 + 1.242·I_L = 3_ − 2.673·I_L = 6_ + 1.431·I_L = 10_ − 0.174·I_W = 1.0_ + 0.174·I_W = 2.0 −_ 0.257·I_L = 3_·I_W = 1.0_ + 0.257·I_L = 3_·I_W = 2.0_ + 0.075·I_L = 6_·I_W = 1.0_ − 0.075·I_L = 6_·I_W = 2.0_ + 0.332·I_L = 10_·I_W = 1.0_ − 0.332·I_L = 10_·I_W = 2.0_(3)*W_ef_*0.1842 + 0.0608·I_L = 3_ − 0.0192·I_L = 6_ − 0.0417·I_L = 10_ + 0.0682·I_W = 1.0_ − 0.0682·I_W = 2.0_ + 0.0188·I_L = 3_·I_W = 1.0_ − 0.0188·I_L = 3_·I_W = 2.0_ − 0.0032·I_L = 6_·I_W = 1.0_ + 0.0032·I_L = 6_·I_W = 2.0_ − 0.0157·I_L = 10_·I_W = 1.0_ + 0.0157·I_L = 10_·I_W = 2.0_(4)

Finally, the main effects plots are provided in the accompanying Figure 11.

Ultimate Tensile Stress

The parameter *σ_u,t_* was highly sensitive to fibre length (L) (*p* = 0.007), whereas wool content (W) did not show a statistically significant effect (*p* = 0.113). The L × W interaction was significant (*p* = 0.024), indicating that the effect of L depends on the level of W. The model explains R^2^ = 58.40% with S = 1.744 MPa, allowing trends to be discriminated with moderate reliability. This pattern is consistent with the means and the main effects plots associated with tension.

Ultimate Flexural Stress

The parameter *σ_u,f_* showed a clear dependence on W (*p* = 0.037), whereas L was not significant (*p* = 0.211). A highly significant L × W interaction was observed (*p* < 0.001), indicating that the magnitude of the effect of wool content varies with fibre length (and vice versa). The model reached R^2^ = 63.11% with S = 5.855 MPa, confirming a well-defined effect structure consistent with the main effects and interaction plots.

Ultimate Compressive Stress

For *σ_u,c_*, L was the only significant factor (*p* = 0.004), whereas W (*p* = 0.731) and the L × W interaction (*p* = 0.887) showed no effects within the evaluated range. The model fit yielded R^2^ = 37.95% with S = 2.738 MPa; despite the higher dispersion typical of this test, the model reliably isolated the length-dependent gradient observed in the main effects plots.

Effective Energy Absorbed

*W_ef_* was governed by W (*p* = 0.001), whereas L showed a marginal contribution (*p* = 0.052) and the L × W interaction was not significant (*p* = 0.710). The model yielded R^2^ = 52.38% with S = 0.0896, reproducing in the main effects plots the systematic reduction in *W_ef_* as the content increased within the studied range.

Overall, the two-factor ANOVA confirmed that the mechanical response depends differentially on fibre length and wool content, depending on the test. In tension, *σ_u,t_* is governed by L, with a significant L × W interaction; *σ_u,f_* depends on W and shows a pronounced L × W interaction; *σ_u,c_* is controlled by L; and for *W_ef_*, the determining factor is W. The adjusted R^2^ values (≈25–55%) and standard errors indicate models adequate for discriminating trends within the studied range. The main effects and interaction plots consistently visualize these patterns in agreement with the *p*-values, showing that short-to-medium fibre lengths and moderate contents maximize performance in a test-dependent manner. These conclusions provide an objective framework for comparing formulations and guiding adjustments of L and W in future developments.

## 4. Conclusions

The mechanical response of wool-reinforced epoxy composites was evaluated as a function of fibre length (3, 6, and 10 mm) and content. The main findings are as follows:1.Tension: The best-performing formulation (10 mm/1.5 wt.%) increased the ultimate tensile stress by ≈10% and tensile modulus by ≈50%, while very short fibres at low content (3 mm/1.5 wt.%) reduced tensile strength by ≈17.5%.2.Flexure: Wool decreased the flexural modulus by ≈15–35%, but selected combinations (3–6 mm/5 wt.%) increased the flexural fracture stress by ≈35–45%.3.Compression: The highest improvement was obtained for 3 mm/2 wt.% fibres, with an increase of ≈160% in compressive yield stress compared with neat epoxy.4.Impact: All wool-filled composites exhibited reduced Charpy impact energy (≈75–93% decrease), with the least severe drop occurring for 3 mm/1 wt.%.

Overall, wool fibres can effectively modify the mechanical response of epoxy, offering improved tensile and compressive properties for selected fibre configurations, with potential use in non-structural or semi-structural components where sustainable materials are preferred.

## Figures and Tables

**Figure 1 materials-18-05391-f001:**
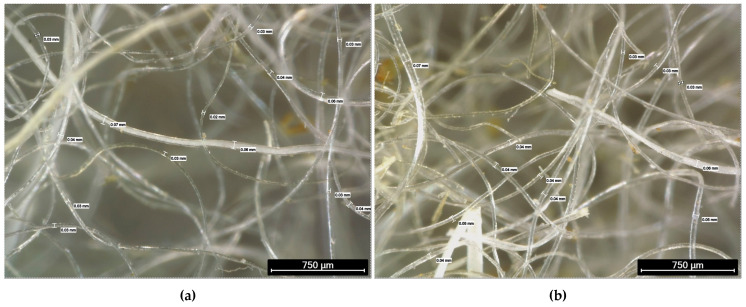
Measurements, appearance, and distribution of Segureña sheep wool reinforcement fibres obtained by optical microscopy: (**a**) light white fibres with low straw residues; (**b**) light white fibres with low straw residues (different field of view); (**c**) white/yellow fibres with low soil/straw residues; (**d**) white/yellow fibres with low soil/straw residues (different field of view).

**Figure 2 materials-18-05391-f002:**
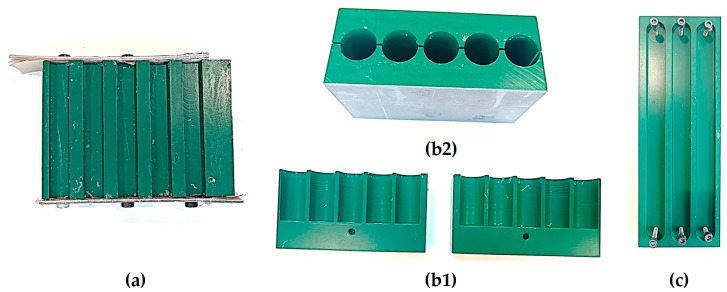
Mechanised moulds: (**a**) flexural and Charpy; (**b1**,**b2**) compression: (**b1**) assembled; (**b2**) halves apart; (**c**) tensile.

**Figure 3 materials-18-05391-f003:**
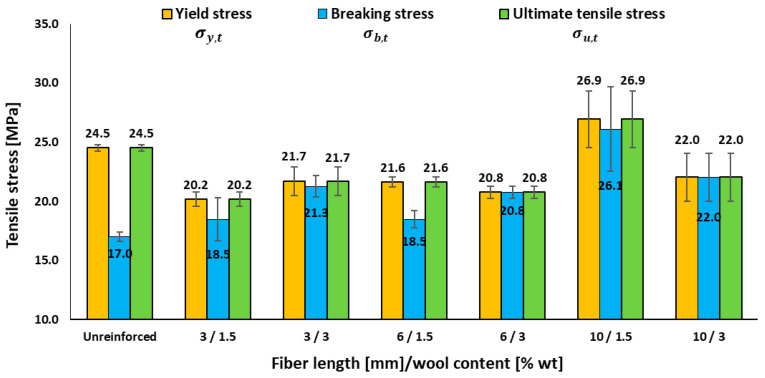
Effects of fibre length and wool content on the tensile performance of epoxy–wool composites: yield stress, breaking stress, and ultimate tensile stress.

**Figure 4 materials-18-05391-f004:**
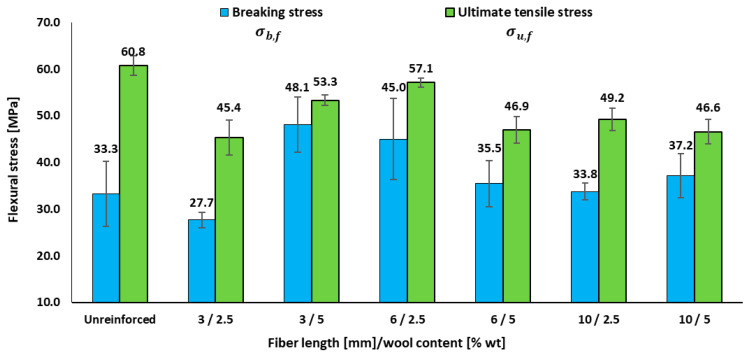
Effects of fibre length and wool content on the flexural performance of epoxy–wool composites: breaking stress and ultimate flexural stress.

**Figure 5 materials-18-05391-f005:**
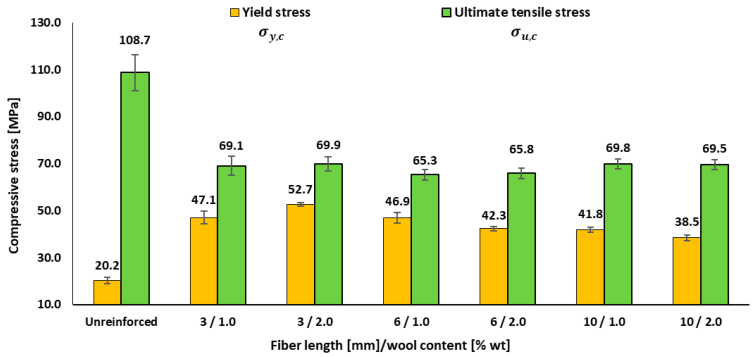
Effects of fibre length and wool content on the compressive performance of epoxy–wool composites: yield stress and ultimate compressive stress.

**Figure 6 materials-18-05391-f006:**
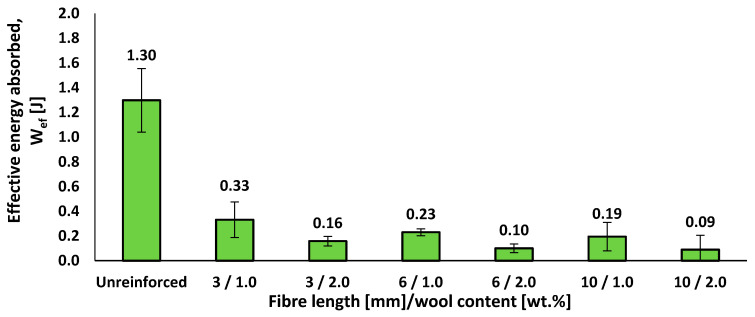
Effective energy absorbed in Charpy impact tests of epoxy–wool composites as a function of fibre length and fibre content.

**Figure 7 materials-18-05391-f007:**
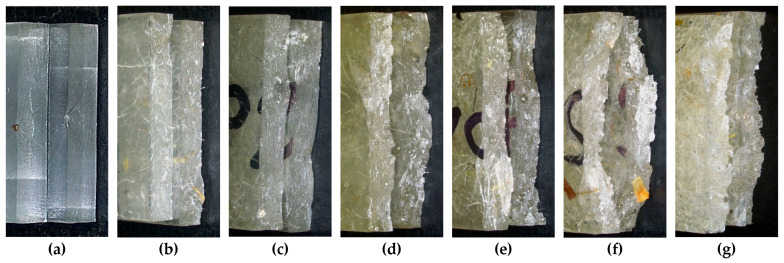
Representative fracture surfaces of specimens after tensile testing (optical micrographs at high magnification obtained with the Leica DVM6-A microscope): (**a**) neat epoxy resin, (**b**) 3 mm fibres/1.5 wt.%, (**c**) 3 mm fibres/3.0 wt.%, (**d**) 6 mm fibres/1.5 wt.%, (**e**) 6 mm fibres/3.0 wt.%, (**f**) 10 mm fibres/1.5 wt.%, and (**g**) 10 mm fibres/3.0 wt.%.

**Figure 8 materials-18-05391-f008:**
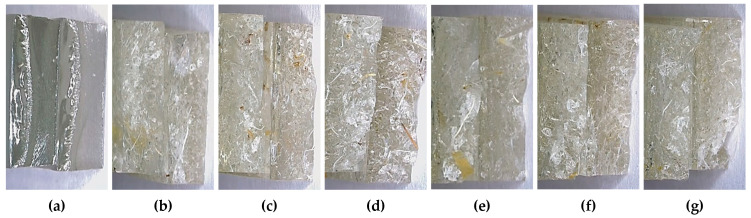
Representative fracture surfaces of specimens after flexural testing (optical micrographs at high magnification obtained with the Leica DVM6-A microscope): (**a**) neat epoxy resin, (**b**) 3 mm fibres/2.5 wt.%, (**c**) 3 mm fibres/5.0 wt.%, (**d**) 6 mm fibres/2.5 wt.%, (**e**) 6 mm fibres/5.0 wt.%, (**f**) 10 mm fibres/2.5 wt.%, and (**g**) 10 mm fibres/5.0 wt.%.

**Figure 9 materials-18-05391-f009:**
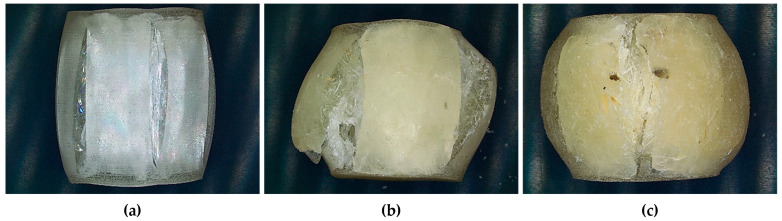
Representative fracture surfaces of specimens after compressive testing: (**a**) neat epoxy resin, (**b**) 3 mm fibres/1 wt.%, and (**c**) 3 mm fibres/2.0 wt.%.

**Figure 10 materials-18-05391-f010:**
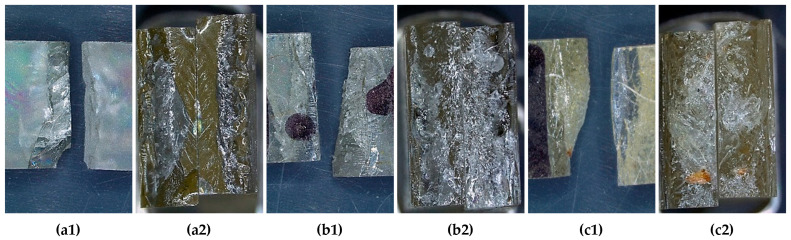
Representative fracture surfaces of specimens after Charpy impact testing (optical images obtained with the Leica DVM6-A microscope): (**a1**) side view of the broken neat epoxy specimen; (**a2**) frontal close-up view of the fracture surface in neat epoxy; (**b1**) side view of the composite with 3 mm wool fibres at 1 wt.% loading; (**b2**) frontal close-up view of the fracture surface for the composite with 3 mm wool fibres at 1 wt.% loading; (**c1**) side view of the composite with 3 mm wool fibres at 2.0 wt.% loading; (**c2**) frontal close-up view of the fracture surface for the composite with 3 mm wool fibres at 2.0 wt.% loading.

**Figure 11 materials-18-05391-f011:**
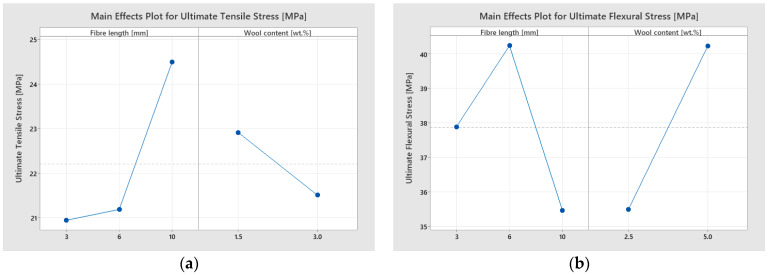
Main effects plots showing the influence of fibre length and wool content on the responses obtained from mechanical tests: (**a**) ultimate tensile stress, (**b**) ultimate flexural stress, (**c**) ultimate compressive stress, and (**d**) effective energy absorbed (impact).

**Table 1 materials-18-05391-t001:** Morphological and visual characterization of Segureña sheep wool fibres, including colour, type, and quantity of residues, along with the fibre diameter measurements obtained by optical microscopy.

Figure	Colour ^1^	Residue Type ^1^	Residue Quantity ^1^	Max. Fibre Diam. (μm)	Min. Fibre Diam. (μm)	Mean Fibre Diam. (μm)
1a	Light white	Straw	Low	70	20	45
1b	Light white	Straw	Low	60	30	45
1c	White/yellow	Soil/straw	Low	60	30	45
1d	White/yellow	Soil/straw	Low	70	20	45

^1^ Values based on subjective visual assessment.

**Table 2 materials-18-05391-t002:** Experimental matrix of moulded–machined specimens.

Test	Fibre Length (mm)	Wool Load (%)	Replicates	Dimensions
Tensile	0	0	3	250 × 25 × 4
	3	1.5		
	3	3.0		
	6	1.5		
	6	3.0		
	10	1.5		
	10	3.0		
Flexural	0	0	5	100 × 10 × 4
	3	2.5		
	3	5.0		
	6	2.5		
	6	5.0		
	10	2.5		
	10	5.0		
Compressive Charpy Impact	0	0	5	Ø20 × 25 ^1^ 10 × 4 × 80 ^2^
	3	1.0		
	3	2.0		
	6	1.0		
	6	2.0		
	10	1.0		
	10	2.0		

^1^ Compressive test dimensions. ^2^ Charpy impact test dimensions.

**Table 3 materials-18-05391-t003:** Tensile properties of epoxy–wool composites (mean ± SD): modulus of elasticity and strain at break.

Specimen Type	Modulus of Elasticity *E_t_* (MPa)			Strain at Break *ε_b,t_* (%)		
Unreinforced	331.0	±	39.2	5.65	±	1.33
3 mm/1.5 wt.%	331.6	±	89.6	0.14	±	0.20
3 mm/3.0 wt.%	282.7	±	20.6	0.00	±	0.00
6 mm/1.5 wt.%	269.2	±	72.7	0.12	±	0.10
6 mm/3.0 wt.%	238.0	±	4.1	0.06	±	0.09
10 mm/1.5 wt.%	490.4	±	101.5	0.19	±	0.20
10 mm/3.0 wt.%	251.0	±	27.3	0.34	±	0.29

**Table 4 materials-18-05391-t004:** Flexural modulus of epoxy–wool composites (mean ± SD) as a function of fibre length and loading.

Specimen Type	Modulus of Elasticity *E_f_* (MPa)		
Unreinforced	1961.3	±	134.5
3 mm/2.5 wt.%	1526.4	±	85.8
3 mm/5.0 wt.%	1672.9	±	45.4
6 mm/2.5 wt.%	1670.4	±	64.6
6 mm/5.0 wt.%	1476.1	±	171.1
10 mm/2.5 wt.%	1580.5	±	85.4
10 mm/5.0 wt.%	1237.2	±	149.9

**Table 5 materials-18-05391-t005:** Compressive properties of epoxy–wool composites (mean ± SD): strain at yield and strain at break as a function of fibre length and loading.

Specimen Type	Strain at Yield *ε_y,c_* (%)			Strain at Break *ε_b,c_* (%)		
Unreinforced	6.6	±	0.5	62.8	±	1.8
3 mm/1.0 wt.%	9.9	±	0.8	42.2	±	16.7
3 mm/2.0 wt.%	10.7	±	1.1	46.3	±	1.4
6 mm/1.0 wt.%	9.7	±	0.3	49.5	±	0.8
6 mm/2.0 wt.%	9.4	±	0.2	50.8	±	1.0
10 mm/1.0 wt.%	9.2	±	0.3	50.3	±	0.9
10 mm/2.0 wt.%	9.5	±	0.4	51.8	±	0.8

**Table 6 materials-18-05391-t006:** Analysis of variance (ANOVA) results for the response variables: *σ_u,t,_ σ_u,f_*, *σ_u,c_*, and *W_ef_*.

Test	Variable	Source	DF	Adj SS	Adj Ms	F-Value	*p*-Value
Tensile	*σ_u,t_*	L (mm)	2	47.237	23.619	7.76	0.007
		W (wt.%)	1	8.904	8.904	2.93	0.113
		L (mm) × W (wt.%)	2	31.658	15.829	5.20	0.024
		Error	12	36.500	3.042		
		Total	17	124.300			
Flexural	*σ_u,f_*	L (mm)	2	113.8	56.89	1.66	0.211
		W (wt.%)	1	167.7	167.7	4.89	0.037
		L (mm) × W (wt.%)	2	1125.9	562.93	16.42	0.000
		Error	24	822.6	34.28		
		Total	29	2229.9			
Compressive	*σ_u,c_*	L (mm)	2	107.360	53.6799	7.16	0.004
		W (wt.%)	1	0.907	0.9071	0.12	0.731
		L (mm) × W (wt.%)	2	1.815	0.9076	0.12	0.887
		Error	24	179.984	7.4994		
		Total	29	290.067			
Charpy Impact	*W_ef_*	L (mm)	2	0.054321	0.027161	3.38	0.052
		W (wt.%)	1	0.128678	0.128678	16.02	0.001
		L (mm) × W (wt.%)	2	0.005576	0.002788	0.35	0.710
		Error	22	0.176660	0.008030		
		Total	27	0.370971			

**Table 7 materials-18-05391-t007:** Summary of the regression models for the response variables: *σ_u,t,_ σ_u,f_*, *σ_u,c_*, and *W_ef_*.

Variable	S	R^2^ [%]
Ultimate Tensile Stress, *σ_u,t_* [MPa]	1.74404	58.40
Ultimate Flexural Stress, *σ_u,f_* [MPa]	5.85461	63.11
Ultimate Compressive Stress, *σ_u,c_* [MPa]	2.73849	37.95
Effective Energy Absorbed, *W_ef_* [J]	0.0896103	52.38

## Data Availability

The original contributions presented in this study are included in the article. Further inquiries can be directed to the corresponding author.

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
