# Peer review of "Mechanical Performance of Wool-Reinforced Epoxy Composites: Tensile, Flexural, Compressive, and Impact Analysis"

_materials, 2025, doi:10.3390/ma18235391_

Round 1

Reviewer 1 Report

Comments and Suggestions for Authors

This study evaluates washed sheep wool fibres as an eco-friendly reinforcement for epoxy composites, analysing their mechanical performance under tensile, flexural, compressive, and impact loading conditions. The main parameters investigated are fibre length and weight fraction. The overall quality and novelty of the research make it suitable for publication in Materials. However, several aspects require further clarification and supporting evidence. For example, in the introduction, the literature review is incomplete, or in the results and discussion the fracture types cannot be identified from the presented microscopic images. Therefore, the manuscript requires major revision before it can be considered for publication. The detailed comments are provided below:

  1. The objective of the work should be highlighted in the abstract. What is the industrial applications for wool reinforced epoxy composite?
  2. Please characterise in the abstract the percentage changes in the mechanical properties
  3. Line 28 to 33, the authors state that “the evolution of the textile industry and the rise of synthetic fibres have drastically reduced the demand for low-grade wool.” Therefore, what is the significance or potential benefit of conducting this study?
  4. In line 46, the authors state “Combining epoxy matrices with natural fibres offers a route to sustainable biocomposites with reduced environmental impact” but Typical epoxy adhesives are not considered biomaterials, or for application of bio composite because they are not biodegradable. Please modify the indicated sentence.
  5. In line 48 to 50, please elaborate in detail the advanced applications of epoxy adhesive in composite structures for example for co-bonding and co-curing of composite parts. Refer to the current recent studies below to describe this.

Mechanical analysis of unidirectional glass fibre reinforced epoxy composite joints manufactured by adhesive bonding and co-curing techniques. Materials & Design, 2025, 114739.

Co-bonding of aerospace composite joints with reflowable adherend interfaces. Journal of Composite Materials, 2025, 00219983251369766.

  1. There is no available literature on short fibre–reinforced epoxy composites that investigates mechanical properties using fibres of similar lengths (3, 6, and 10 mm) and contents (1.0–5.0 wt.%). Such studies are necessary to enable a comparison of the mechanical properties of these composites with those of the wool–epoxy composite.
  2. Please describe the preparation procedure for the standard samples, including their dimensions and layup process.
  3. In Fig. 3, could you elaborate on why the addition of 1.5 wt% wool fibres with a length of 3 mm resulted in a 17.5% decrease in ultimate tensile strength compared to neat epoxy? It appears that the fibre length might be too large. This is also observed with 25.3% decreases in flexural strength (Fig.4)
  4. Please include high-magnification microscopic images of the fracture surface after the tensile test to show the fracture type (fibre breakage or fibre pull-out). The fracture type cannot be identified in Fig. 8.

Author Response

Comments 1: The objective of the work should be highlighted in the abstract. What is the industrial applications for wool reinforced epoxy composite?

Response 1: Thank you for this valuable comment. We agree with the reviewer and have revised the abstract accordingly. Specifically, we have (i) clearly stated the main objective of the study and (ii) added a sentence describing the potential industrial applications of wool-reinforced epoxy composites. These changes are marked using Track Changes.

The objective of this work is to assess whether short, washed sheep wool fibres can function as a sustainable reinforcement for epoxy matrices and to identify optimal fibre length–content windows that improve mechanical behaviour for engineering applications.”

“…pointing to their use in non-structural to semi-structural industrial components such as interior panels, housings, casings, protective covers and other parts where moderate tensile/compressive performance is sufficient and material sustainability is prioritised.”

Comments 2: Please characterise in the abstract the percentage changes in the mechanical properties.

Response 2: Thank you for this valuable suggestion. We agree with the reviewer and have revised the abstract to include quantitative percentage changes for the tensile, flexural, compressive, and impact properties. These changes are marked using Track Changes.

“…corresponding to gains of about 10% in ultimate tensile stress and 50% in tensile modulus, at the expense of ductility. In flexure, the modulus decreases by roughly 15–35% compared with the matrix, whereas configurations with 3–6 mm at 2.5-5 wt.% raise fracture stress by about 35–45% and improve post-peak resistance . In compression, reinforcement markedly elevates yield stress, with increases up to about 160% at 3 mm/2 wt.%, while ultimate strain decreases moderately. In Charpy impact, all reinforced materials underperform the resin, with absorbed energy reduced by roughly 75–93% depending on fibre length and content, 3 mm/1 wt.% being least affected.”

Comments 3: Line 28 to 33, the authors state that “the evolution of the textile industry and the rise of synthetic fibres have drastically reduced the demand for low-grade wool.” Therefore, what is the significance or potential benefit of conducting this study?

Response 3: Thank you for this comment. We agree that the significance of using low-grade wool needed to be clarified. We have therefore revised the Introduction to explain that most sheep producing coarse or heterogeneous wool are primarily farmed for meat, milk, or grazing management rather than for fibre. These animals must still be shorn annually for health and welfare reasons, which generates a continuous and unavoidable supply of low-grade wool with little or no commercial value. Highlighting this aspect reinforces the relevance of the study, as it shows that valorising this by-product as reinforcement in epoxy composites provides both environmental and economic benefits. These additions are marked with Track Changes in the revised manuscript.

Most of the flocks that generate this coarse, heterogeneous wool are primarily farmed for meat, milk, or grazing management rather than for fibre, but the animals must still be shorn annually for health and welfare reasons, so low-grade wool continues to be produced as an unavoidable by-product with little or no market value.”

“…and motivates the present study on their use as reinforcement in epoxy composites.”

Comments 4: In line 46, the authors state “Combining epoxy matrices with natural fibres offers a route to sustainable biocomposites with reduced environmental impact” but Typical epoxy adhesives are not considered biomaterials, or for application of bio composite because they are not biodegradable. Please modify the indicated sentence.

Response 4: Thank you for this observation. We agree that conventional epoxy matrices are not biomaterials and are not biodegradable. To avoid overstatement, we have revised the sentence in the Introduction to clarify that the potential environmental benefit arises from the incorporation of renewable natural fibres and improved life-cycle metrics, while explicitly acknowledging that epoxy itself is a non-biodegradable polymer. The updated wording is marked with Track Changes in the revised manuscript.

Combining epoxy matrices with natural fibres has been proposed as a route towards composites with higher renewable content and improved life-cycle metrics, even though conventional epoxy matrices are not biopolymers and are not biodegradable.”

Comments 5: In line 48 to 50, please elaborate in detail the advanced applications of epoxy adhesive in composite structures for example for co-bonding and co-curing of composite parts. Refer to the current recent studies.

Response 5: Thank you for this helpful suggestion. We have revised the Introduction to clarify that epoxy resins are not only widely used as matrix materials but also as structural adhesives in advanced composite joints, including bonded, co-bonded, and co-cured connections in aerospace, automotive, and marine structures. The revised text now explicitly mentions these applications and cites the suggested recent studies on adhesively bonded and co-cured glass-fibre epoxy joints and on co-bonded aerospace composite joints. These modifications are marked with Track Changes in the revised manuscript.

Epoxy resins, in turn, are staple thermosets for advanced composite structures because of their strong adhesion, chemical resistance, and thermal stability, and they are widely used both as matrix resins and as structural adhesives in bonded, co-bonded, and co-cured composite joints in aerospace, automotive, and marine applications.”

Comments 1b: There is no available literature on short fibre–reinforced epoxy composites that investigates mechanical properties using fibres of similar lengths (3, 6, and 10 mm) and contents (1.0–5.0 wt.%). Such studies are necessary to enable a comparison of the mechanical properties of these composites with those of the wool–epoxy composite.

Response 1b: We appreciate this remark. We agree that there is no directly comparable literature on short-fibre epoxy composites using the same combination of fibre lengths (3, 6, and 10 mm) and contents (1.0–5.0 wt.%) adopted in this work. To clarify this point, we have added a statement at the end of the Introduction explaining that, to the best of our knowledge, no unified study exists in this specific length–content window, and that our mechanical results can therefore serve as a reference for future comparisons with other short-fibre epoxy systems. The new text is marked with Track Changes in the revised manuscript.

“To the authors’ knowledge, there is no available study on short-fibre–reinforced epoxy composites that systematically covers fibre lengths of 3, 6, and 10 mm combined with contents of 1.0–5.0 wt.% under a unified experimental design. This lack of directly comparable data underscores the need for reference mechanical results in this length–content window to benchmark wool–epoxy systems against other short-fibre composites.”

Comments 2b: Please describe the preparation procedure for the standard samples, including their dimensions and layup process.

Response 2b: We thank the reviewer for this comment. We have expanded Section 2.3 (Methods) to describe the fabrication of the moulded–machined specimens in more detail, including the gravity-casting procedure, the curing conditions, and the use of short, randomly oriented wool fibres (no laminate lay-up). We now explicitly refer to the standardized specimen dimensions reported in Table 2. These modifications are marked with Track Changes in the revised manuscript.

Standard specimens for tensile, flexural, compressive and Charpy impact tests were produced by gravity casting of wool–epoxy mixtures into aluminium moulds followed by machining to the final geometries specified in Table 2. The two epoxy components were weighed at a 2:1 resin:hardener mass ratio and hand-mixed for approximately 5 min. Washed Segureña wool was cut to nominal fibre lengths of 3, 6 or 10 mm and weighed to reach the target mass fractions (1.0–5.0 wt.%). For each formulation, the required amount of wool was gradually incorporated into the fresh resin under manual stirring until a visually homogeneous suspension of short, randomly oriented fibres was obtained (no laminates or stacking sequence were used). The mixtures were then poured by gravity into FEP-coated aluminium moulds: flat plate cavities for tensile, flexural and Charpy bars, and cylindrical cavities for compression specimens. To limit air entrapment, mixing was performed manually at low speed and the moulds were filled slowly from one side, allowing bubbles to rise to the free surface before gelation. No additional vacuum-degassing step was applied in this study. After initial gelation at room temperature, the filled moulds were cured in a ventilated oven at 60 °C for 24 h, demoulded, and finally machined to the standardized dimensions listed in Table 2.”

Comments 3b: In Fig. 3, could you elaborate on why the addition of 1.5 wt% wool fibres with a length of 3 mm resulted in a 17.5% decrease in ultimate tensile strength compared to neat epoxy? It appears that the fibre length might be too large. This is also observed with 25.3% decreases in flexural strength (Fig.4).

Response 3b: We thank the reviewer for this observation. We have expanded the discussion in Sections 3.1 and 3.2 to explain why the formulation with 3 mm fibres at 1.5 wt.% shows a 17.5% decrease in ultimate tensile strength and up to approximately 25% reduction in flexural strength compared with the neat epoxy. The corresponding explanations have been incorporated into the manuscript and are marked with Track Changes.

“It is worth noting that, for the shortest fibre length at the lowest content (3 mm / 1.5 wt.%), the ultimate tensile stress decreases by about 17.5% compared with the neat epoxy. This behaviour is consistent with a defect-dominated regime in which fibres that are below or close to the effective load-bearing length do not contribute signifi-cantly to tensile reinforcement. Instead, they primarily act as additional stress concen-trators (fibre ends, local stiffness mismatches, voids and interfacial defects), promoting earlier crack initiation and limiting the potential for fibre bridging and pull-out to in-crease the load-carrying capacity.”

“For flexural loading, the shortest wool fibres also lead to a reduction in strength. The incorporation of 3 mm fibres produces decreases in flexural strength of up to ap-proximately 25% relative to the neat resin, depending on fibre content, as illustrated in Figure 4. As in tension, this behaviour is consistent with a defect-dominated regime, where very short fibres mainly introduce additional stress concentrators (fibre ends, local stiffness mismatches, voids and interfacial decohesion) in the tensile zone of the bent specimens, instead of acting as effective load-bearing bridges. This interpretation agrees with the fracture morphologies observed around short fibres (Section 3.5).”

Comments 4b: 1.         Please include high-magnification microscopic images of the fracture surface after the tensile test to show the fracture type (fibre breakage or fibre pull-out). The fracture type cannot be identified in Fig. 8.

Response 4b: We thank the reviewer for this valuable comment. The method used to examine the fracture morphology has now been described in greater detail in Sections 2.2 and 3.5. In the revised manuscript, we clarify the optical configuration of the Leica DVM6-A digital microscope employed for all fractographic observations (reflected-light mode, FOV 43.75 objective, 1600 × 1200 px, spatial sampling ≈ 21–22 μm/px). This information specifies the magnification level, spatial resolution, and imaging conditions under which the fracture surfaces were assessed.

Considering the typical diameter of wool fibres (20–40 μm), the selected optical configuration provides sufficient magnification and resolution to distinguish the fracture features relevant to this study—including fibre pull-out, interfacial debonding, and matrix cracking—without requiring SEM analysis. These fracture mechanisms were clearly identified under the high-magnification optical imaging conditions used and are now explicitly described and discussed in the revised Section 3.5.

The new text added to Section 2.2 and Section 3.5 (shown in Track Changes in the revised manuscript) clarifies these aspects.

“Fractographic observations were performed with a Leica DVM6-A digital microscope operated in reflected-light mode under the optical configuration described in Section 2.2 (FOV 43.75 objective, 1600 × 1200 px, spatial sampling ≈ 21–22 μm/px), which in this study allowed us to visually distinguish individual wool fibres (≈20–40 μm in diameter) and to qualitatively identify their failure mode (predominantly fibre pull-out rather than fibre breakage). (…) In all reinforced tensile specimens, most wool fibres fail by interfacial debonding followed by pull-out, whereas clean fibre breakage is comparatively rare. This pull-out-dominated fracture mode is consistent with the modest increase in ultimate tensile stress and the severe loss of ductility reported in Section 3.1.”

Reviewer 2 Report

Comments and Suggestions for Authors

1 The manuscript by Carlos et al investigated the mechanical properties of wool reinforced epoxy composites. This paper should be the first in the literature to report such an interesting topic using wool as reinforcement and epoxy resin as matrix, and the work is innovative.

2 In this paper, the mechanical properties of wool/epoxy resin composites were investigated. The tensile, flexural, compressive and impact properties of the composites were explored. The effects of the length and dosage of the fiber on the mechanical properties of the composites were comparatively studied, and some conclusions were draw for each property. The findings in this article have certain reference values for other researchers who are interested in this top, such as the epoxy value of the resin and the kind of the hardener.

3 In line 143, the loading velocity for tensile test was defined as 0.11–0.20 mm/s, as we know, the velocity has great effect on the tensile results, no certain tensile velocity was not acceptable for a comparative test, Generally, the result with a crosshead speed of 0.11 mm/s would differ from that of 0.20 mm/s. If the test was finished using different tensile velocities, the reason should be explained. If only one velocity was adopted, it should be described definitely.

4 In Table 2, the sizes of the specimen for the flexural test was 10 × 4 × 80, while the dimensions shown in line 149 for flexural test were 100 × 10 × 4 mm, there existed great differences between the dimensions of the actual sample and the model, why? 

5 the method to exam the fracture morphology has not been described clearly though it was mentioned in line 163. For mechanical performance analysis, SEM observations are often needed to correlate the micro structure and macro properties of the samples.

6 the preparation methods for the samples should be introduced in detailed, especially how to degas during the preparation should be described clearly, as we know, the viscosity of epoxy resin is heavy, much air would be introduced into the sample during its preparation if the operation was not controlled properly. The incorporation of air and its content would affect the mechanical performances seriously.

7 the mechanical results have been demonstrated clearly, but the mechanism has not been explored enough. Some theoretical analyses should be emphasized in the paper.

8 The “4. Conclusions” is too long and dispersive. Rewrite it again trying to be more incisive on the content of the article.

9 All the references are appropriate.

Author Response

Comments 3: In line 143, the loading velocity for tensile test was defined as 0.11–0.20 mm/s, as we know, the velocity has great effect on the tensile results, no certain tensile velocity was not acceptable for a comparative test, Generally, the result with a crosshead speed of 0.11 mm/s would differ from that of 0.20 mm/s. If the test was finished using different tensile velocities, the reason should be explained. If only one velocity was adopted, it should be described definitely.

Response 3: We thank the reviewer for pointing out this ambiguity. In the revised manuscript we now clarify that a single crosshead speed was used for all tensile tests. Although the original equipment settings displayed the allowable operating range of 0.11–0.20 mm/s, the actual value applied during all experiments was 0.20 mm/s, as fixed in the testing machine prior to sample characterization. This information has been added in Section 3.2.1 / 3.3 (Tensile test procedure).

Comments 4: In Table 2, the sizes of the specimen for the flexural test was 10 × 4 × 80, while the dimensions shown in line 149 for flexural test were 100 × 10 × 4 mm, there existed great differences between the dimensions of the actual sample and the model, why? 

Response 4: We thank the reviewer for pointing out this inconsistency. The correct dimensions of the flexural specimens are 100 × 10 × 4 mm, following UNE-EN ISO 178, as indicated in line 149. The value reported in Table 2 (10 × 4 × 80 mm) was the result of an editing error. The table has now been corrected to match the actual specimen dimensions used in the experimente.

Comments 5: The method to exam the fracture morphology has not been described clearly though it was mentioned in line 163. For mechanical performance analysis, SEM observations are often needed to correlate the micro structure and macro properties of the samples.

Response 2: We thank the reviewer for this observation. The method used to examine the fracture morphology has now been described more clearly in Sections 2.2 and 3.5. In the revised manuscript, we specify the optical configuration of the Leica DVM6-A digital microscope used for all fractographic analyses (reflected-light mode, FOV 43.75 objective, 1600 × 1200 px, spatial sampling ≈ 21–22 μm/px). This information clarifies the magnification, resolution and imaging conditions under which the fracture surfaces were evaluated.

Given the diameter of wool fibres (20–40 μm), this optical configuration provides sufficient spatial resolution to identify the relevant fracture mechanisms—such as fibre pull-out, interfacial debonding, and matrix cracking—without requiring SEM observations. The dominant fracture modes identified in the revised manuscript were clearly observed with the optical microscope configuration used in this study and have been incorporated into Section 3.5. The new text added to Section 2.2 and Section 3.5 reads as follows (Track Changes in the revised manuscript):

“For fracture-surface inspection, the microscope was operated in reflected-light mode with an FOV 43.75 objective (nominal NA = 0.007). Images were acquired at 1600 × 1200 px (8-bit RGB), corresponding to a field of view of approximately 34.5 × 25.8 mm and a spatial sampling of ≈ 21–22 μm per pixel.”

Comments 6: The preparation methods for the samples should be introduced in detailed, especially how to degas during the preparation should be described clearly, as we know, the viscosity of epoxy resin is heavy, much air would be introduced into the sample during its preparation if the operation was not controlled properly. The incorporation of air and its content would affect the mechanical performances seriously.

Response 6: We thank the reviewer for highlighting this important point. We have expanded the description of the specimen preparation procedure in Section 2.3. The revised text now details the mixing sequence, fibre incorporation and casting procedure, and it explicitly addresses how air entrapment was controlled.

Standard specimens for tensile, flexural, compressive and Charpy impact tests were produced by gravity casting of wool–epoxy mixtures into aluminium moulds followed by machining to the final geometries specified in Table 2. The two epoxy components were weighed at a 2:1 resin:hardener mass ratio and hand-mixed for approximately 5 min. Washed Segureña wool was cut to nominal fibre lengths of 3, 6 or 10 mm and weighed to reach the target mass fractions (1.0–5.0 wt.%). For each formulation, the required amount of wool was gradually incorporated into the fresh resin under manual stirring until a visually homogeneous suspension of short, randomly oriented fibres was obtained (no laminates or stacking sequence were used). The mixtures were then poured by gravity into FEP-coated aluminium moulds: flat plate cavities for tensile, flexural and Charpy bars, and cylindrical cavities for compression specimens. To limit air entrapment, mixing was performed manually at low speed and the moulds were filled slowly from one side, allowing bubbles to rise to the free surface before gelation. No additional vacuum-degassing step was applied in this study. After initial gelation at room temperature, the filled moulds were cured in a ventilated oven at 60 °C for 24 h, demoulded, and finally machined to the standardized dimensions listed in Table 2.”

Comments 7: The mechanical results have been demonstrated clearly, but the mechanism has not been explored enough. Some theoretical analyses should be emphasized in the paper.

Response 7: We thank the reviewer for this valuable suggestion. In the revised manuscript, the mechanistic interpretation has been strengthened in the sections dealing with tensile and flexural behaviour, with additional discussion of the underlying micromechanical mechanisms and their theoretical background incorporated in Sections 3.1, 3.2 and 3.5.

“It is worth noting that, for the shortest fibre length at the lowest content (3 mm / 1.5 wt.%), the ultimate tensile stress decreases by about 17.5% compared with the neat epoxy. This behaviour is consistent with a defect-dominated regime in which fibres that are below or close to the effective load-bearing length do not contribute signifi-cantly to tensile reinforcement. Instead, they primarily act as additional stress concen-trators (fibre ends, local stiffness mismatches, voids and interfacial defects), promoting earlier crack initiation and limiting the potential for fibre bridging and pull-out to in-crease the load-carrying capacity.”

“For flexural loading, the shortest wool fibres also lead to a reduction in strength. The incorporation of 3 mm fibres produces decreases in flexural strength of up to ap-proximately 25% relative to the neat resin, depending on fibre content, as illustrated in Figure 4. As in tension, this behaviour is consistent with a defect-dominated regime, where very short fibres mainly introduce additional stress concentrators (fibre ends, local stiffness mismatches, voids and interfacial decohesion) in the tensile zone of the bent specimens, instead of acting as effective load-bearing bridges. This interpretation agrees with the fracture morphologies observed around short fibres (Section 3.5).”

In all reinforced tensile specimens, most wool fibres fail by interfacial debonding followed by pull-out, whereas clean fibre breakage is comparatively rare. This pull-out-dominated fracture mode is consistent with the modest increase in ultimate tensile stress and the severe loss of ductility reported in Section 3.1.”
“As in tension, the dominant fracture mechanisms are fibre pull-out and interfacial debonding rather than complete fibre rupture, so flexural energy dissipation is mainly governed by crack deflection around partially debonded fibres.”

Comments 8: The “4. Conclusions” is too long and dispersive. Rewrite it again trying to be more incisive on the content of the article.

Response 8: We thank the reviewer for this helpful observation. In the revised manuscript, Section 4 (Conclusions) has been completely rewritten to be much more concise and focused strictly on the main quantitative findings of the study. The new version summarises the key tensile, flexural, compressive and impact results in a short, structured format, removing all previously redundant or descriptive content. The updated conclusions are now sharply aligned with the core outcomes of the work, as recommended. All modifications are clearly marked with Track Changes in the revised manuscript.

The mechanical response of wool–reinforced epoxy composites was evaluated as a function of fibre length (3, 6, 10 mm) and content. The main findings are::

1.        Tension: the best-performing formulation (10 mm / 1.5 wt.%) increased ultimate tensile stress by ≈10% and tensile modulus by ≈50%, while very short fibres at low content (3 mm / 1.5 wt.%) reduced tensile strength by ≈17.5%..

2.        Flexure: Wool decreased the flexural modulus by ≈15–35%, but selected combinations (3–6 mm / 5 wt.%) increased flexural fracture stress by ≈35–45%.

3.        Compression: The highest improvement was obtained for 3 mm / 2 wt.% fibres, with an increase of ≈160% in compressive yield stress compared with neat epoxy.

4.        Impact: All wool-filled composites exhibited reduced Charpy impact energy (≈75–93% decrease), with the least severe drop occurring for 3 mm / 1 wt.%.

Overall, wool fibres can effectively modify the mechanical response of epoxy, offering improved tensile and compressive properties for selected fibre configurations, with potential use in non-structural or semi-structural components where sustainable materials are preferred.”

Round 2

Reviewer 1 Report

Comments and Suggestions for Authors

Accept in present form